

# Person image generation through graph-based and appearance-decomposed generative adversarial network

Yuling He[1,*], Yingding Zhao[2,*], Wenji Yang[2] and Yilu Xu[2]

[1] School of Computer and Information Engineering, Jiangxi Agricultural University, NanChang, JiangXi, China
[2] School of Software, Jiangxi Agricultural University, NanChang, JiangXi, China
* These authors contributed equally to this work.

## ABSTRACT

Due to the sophisticated entanglements for non-rigid deformation, generating person images from source pose to target pose is a challenging work. In this paper, we present a novel framework to generate person images with shape consistency and appearance consistency. The proposed framework leverages the graph network to infer the global relationship of source pose and target pose in a graph for better pose transfer. Moreover, we decompose the source image into different attributes (*e.g.*, hair, clothes, pants and shoes) and combine them with the pose coding through operation method to generate a more realistic person image. We adopt an alternate updating strategy to promote mutual guidance between pose modules and appearance modules for better person image quality. Qualitative and quantitative experiments were carried out on the DeepFashion dateset. The efficacy of the presented framework are verified.

# INTRODUCTION

Recently generating human images from source pose to target pose, which is commonly known as pose transfer, have obtained great attention and are of great value in many tasks such as intelligent photo editing (*Wu, Xu & Hall, 2017*), film production (*Cui & Wang, 2019*; *Xiong et al., 2018*), virtual try-on (*Dong et al., 2019*; *Honda, 2019*; *Kubo, Iwasawa & Matsuo, 2018*) and person re-identification (*Alqahtani, Kavakli-Thorne & Liu, 2019*; *Dai et al., 2018*; *Liu et al., 2019*; *Lv & Wang, 2018*). *Ma et al. (2017)* first proposed this problem, where the framework transformed person images to target pose while keeping the appearance details of the source image. Then more researchers put forward pose transfer networks (*Huang et al., 2020*; *Song et al., 2019*; *Tang et al., 2020*; *Zhu et al., 2019*). The methods mentioned above are all based on convolutional neural network, which is good at extracting local relations, but it is inefficient in dealing with global interring regional relations. To obtain a greater receptive field, the traditional CNN needs to stack many layers. This may cause some problems of obtaining the global relationship between source pose and target pose, which not only increases the calculation cost, but also has incomplete problems.

Corresponding author
Wenji Yang, ywenji614@jxau.edu.cn

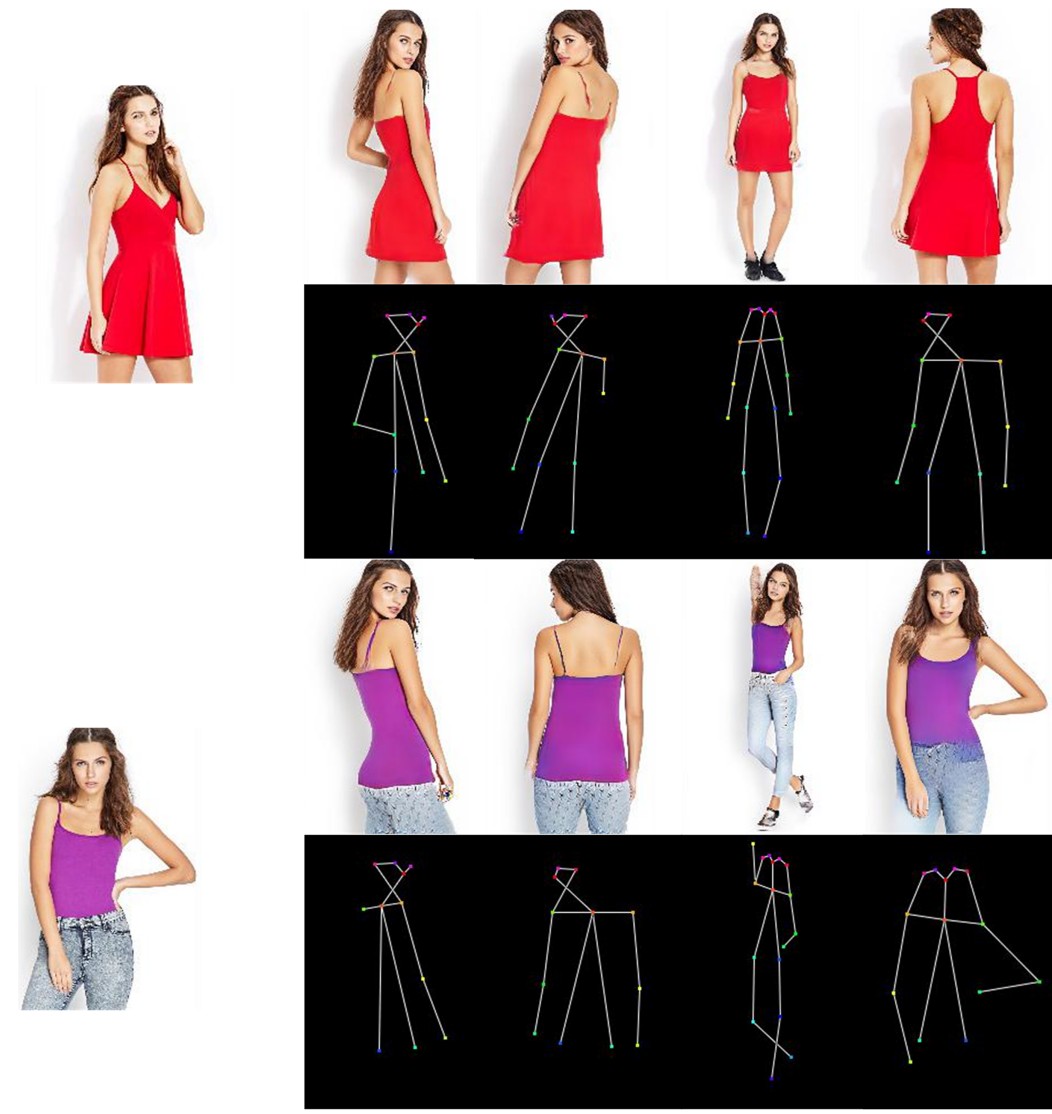

**Figure 1 The results of our method in the pose transfer task.**

   As shown in Fig. 1, in this paper, we propose a pose transfer framework based on graph network and appearance decomposition. Through the graph network, we make a global reasoning between the target pose and the source pose, and get a better attitude transfer effect. Through the appearance attribute decomposition module, the generated person image can obtain more real and delicate appearance features. Inspired by *Chen et al. (2019)*, we map the source pose and target pose to the same interaction space. After global reasoning in the interaction space, we map the different poses back to the original independent space. Specifically, as shown in Fig. 2, we construct an interaction space for global reasoning, map the key points of the source pose and the target pose to the interaction space respectively, establish a fully connected graph connecting all the joint points in the space, and carry out relationship reasoning on the graph. After reasoning is

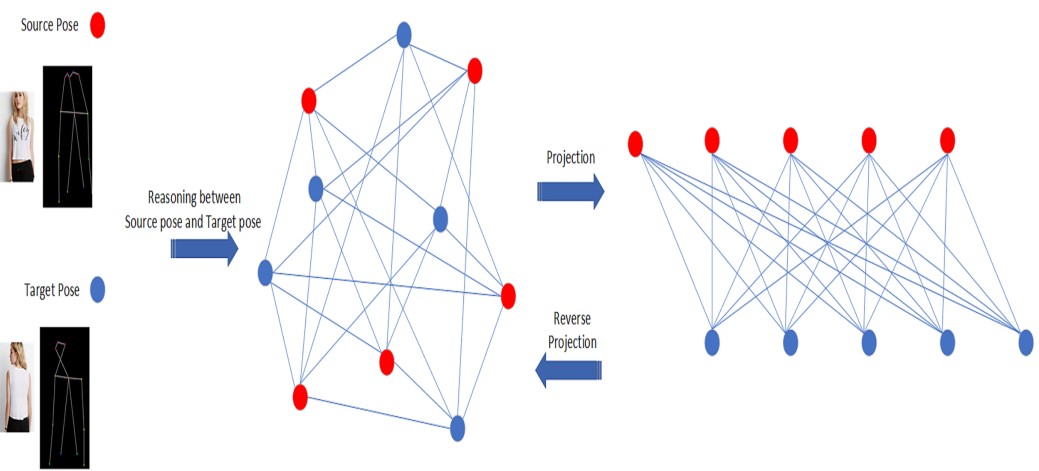

**Figure 2 Illustration of our idea.** The red and blue nodes represent the source pose and the target pose respectively (not all the key points are shown in the figure for convenience). The nodes are mapped from the original space to the interactive space to form a connected graph for reasoning. Then the nodes are projected back to the original space for further processing.

completed, the updated joint points are remapped back to the original space. For appearance code, we use a VGG-based pre-trained human parser to decompose the attributes of source images. Then these attributes are input into a texture encoder to reconstruct the style code, and finally the style code and the pose code are combined to obtain the generated images. In the training process, we use a pair of conditional discriminators, which combine the conditional discriminator and appearance discriminator to improve the quality of the generated image. The performance of proposed network outperform prior works both qualitatively and quantitatively on challenging benchmarks. In total, the proposed framework has the following contributions:

- We propose a novel generative adversarial network based on graph, which can infer the global relationship between different pose. Tackling the problem that CNN needs to overlay multiple convolution layers to expand the receptive field to cover all the joint points of source pose and target pose.
- We employ the human body parser to decompose the attributes of the human body images, and fuse the attribute coding with the pose coding. Therefore, the generated images are desirable.

The remainder of this paper is structured as follows. In "Related Work", the related work of this paper is introduced. "Methods" details of the proposed framework are given. "Results and Discussion" presents distinct experiments on DeepFashion dataset. Finally, a summary is given in the "Conclusion".

## RELATED WORK

### Person image generation

With the continuous development of computer vision technology, image generation models have been developing at a high rate in recent years. The two mainstream methods

are Variational auto-encoder (VAE) (*Kingma & Welling, 2013*; *Lassner, Pons-Moll & Gehler, 2017*; *Rezende, Mohamed & Wierstra, 2014*; *Sohn, Lee & Yan, 2015*) and Generative Adversarial Networks (GANs) (*Balakrishnan et al., 2018*; *Dong et al., 2019*; *Honda, 2019*; *Si et al., 2018*; *Zanfir et al., 2018*). The former captures the relationship between different dimensions of the data by modeling the structure of the data to generate new data. The latter generates images through mutual game between the generator and the discriminator. Since the loss used by GANs is better than VAE, GANs can generate more vivid images and is sought after by more researchers.

Aiming at the human body image generation method based on the generative adversarial network, *Ma et al. (2017)* first proposed PG$^2$ to achieve pose guided person body image generation, whose model is cascaded by two different generators. The first stage generates a blurry image under the target pose. The second stage improve the texture and color quality of the image generated in the first stage. Although the second stage improve the image quality to a certain extent, it is still unable to capture the changes in image distribution well, which makes the generated images lack of fine texture. To obtain better appearance texture (*Esser, Sutter & Br, 2018*), exploited to combine VAE and U-Net to disentangle appearance and pose, using the decoupled posture information to generate pictures, and then integrate the appearance information of the source images into the generated pictures. However, it will cause the problem of feature offset caused by posture difference due to the U-Net based skip connections in the model. To tackle this problem (*Siarohin et al., 2018*), introduced deformable skip connection to transfer features of various parts of the body, which effectively alleviated the problem of feature migration. In order to control the attributes flexibly (*Men et al., 2020*), proposed Attribute-Decomposed GAN, which embeds the attribute codes of each part of the human body into the potential space independently, and recombines these codes in a specific order to form a complete appearance code, so as to achieve the effect of flexible control of each attribute.

## Graph-based reasoning

Graph is a data structure, which can model a group of objects (nodes) and their relationships (edges). In recent years, more and more attention has been paid to the study of graph analysis based on machine learning due to its powerful expression ability. *Kipf & Welling (2016)* firstly proposed graph convolutional network which used an efficient layer-wise propagation rule that is based on a first-order approximation of spectral convolutions on graphs. In order to pay dynamical attention to the features of adjacent nodes, Graph attention networks (*Veliikoviç et al., 2017*) have been proposed. *Wang, Huang & Wang (2020)* introduced a Global Relation Reasoning Graph Convolutional Networks (GRR-GCN) to efficiently capture the global relations among different body joints. It modeled the relations among different body joints that may mitigate some challenges such as occlusion. In this paper, we introduce a graph-based reasoning in person image generation model.

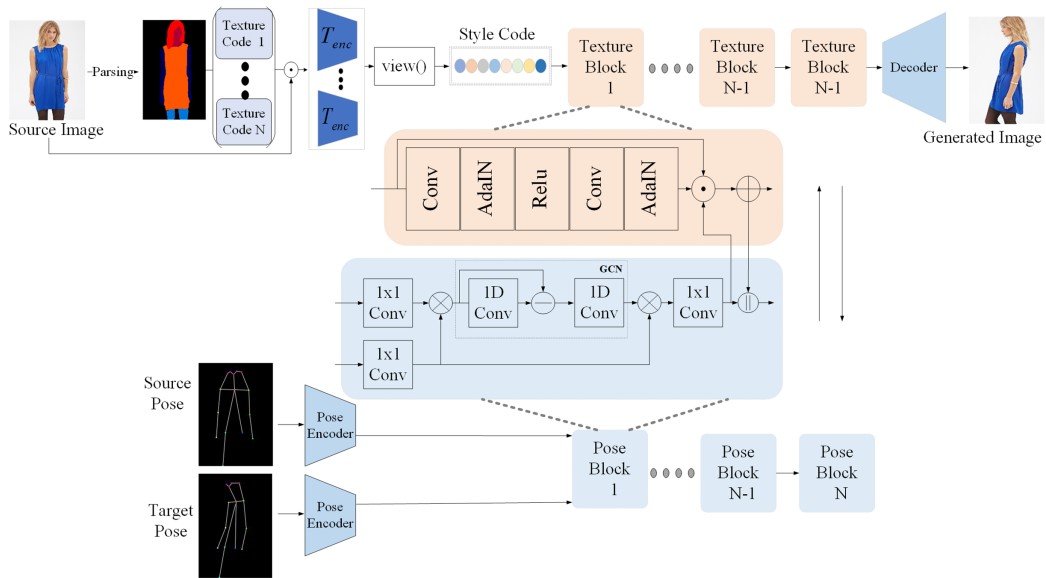

**Figure 3 Structure of our proposed method.**

## METHODS

In this section, we give a description of our network architecture. We start with some notations. $I \in R^{3 \times H \times W}$ denotes the set of person images. Before training, the Human Pose Estimator (HPE) (*Esser, Sutter & Br, 2018*) is adopted to estimate the position of 18 joint points in the images. $P \in R^{18 \times H \times W}$ represents a 18-channel heat map that encodes the locations of 18 joints of a human body. During the training, the model requires source images and target images $(I_c, I_t)$ and their corresponding heat map $(P_c, P_t)$ as input. Moreover, we adopt a VGG-based pre-trained human parser to decompose the attributes of source images. More details will be introduced below.

### Generator

Figure 3 shows the architecture of the generator which aims to transfer the pose of the person in $I_c$ from $P_c$ to $P_t$. At the core the generator comprise two pathway, namely pose pathway and appearance pathway. The former consisted of a series of pose blocks and the latter consisted of several texture blocks.

### *Style encoder*

Due to the manifold structure composed of various human body images is very complex, it is difficult to encode the entire human body with detailed textures. Inspired by *Men et al. (2020)*, we decompose the source image into different components and recombine their potential code to build the complete style code. Firstly, a pre-trained human body parser based on VGG is used to obtain the semantic map of the source image $I_c$. Then, the semantic map is mapped to a $K$-channels heat map $M^{K \times H \times W}$. Each channel $i$ has a binary mask $M_i^{H \times W}$ corresponding to different components. Multiplying element-wise the

source image $I_c$ and the mask $M_i^{H \times W}$ to obtain decomposed person image with component $i$.

$$I_c^i = I_c \odot M_i \tag{1}$$

After that, $I_c^i$ is input into the appearance encoder to acquire the corresponding style code $F_{sty}^i$.

$$F_{sty}^i = T_{enc}(I_c^i) \tag{2}$$

where $T_{enc}$ is shared for all components and then all $F_{sty}^i$ is concatenated to get a full style code $F_{sty}$. The structure of $T_{enc}$ adopts the encoder in *Men et al. (2020)*.

### Pose encoder

In the pose pathway, the source pose $P_c$ and target pose $P_t$ are embedded into the latent space as the pose code $F^{P_c}$ and $F^{P_t}$ by pose encoder. Note that we adopt the same shape encoder for $P_c$ and $P_t$ which consists of $N$-down-sampling convolutional layers ($N = 2$ in our case). That is to say two shape encoders are sharing the weights.

### Pose block

The pose block aims to reason the crossing long range relations between the source pose and the target pose in a graph and output new shape codes. The main idea of this method are to map the source pose and the target pose to the graph space, then cross reasoning on the graph space, and finally map back to the original space to get the updated code. Firstly, we learn the projection function that maps source pose and target pose from coordinating space to graph space.

$$H_{\text{source}} = \theta(F^{P_c}) \in \mathbb{R}^{C \times D} \tag{3}$$
$$H_{\text{target}} = \theta(F^{P_t}) \in \mathbb{R}^{C \times D} \tag{4}$$

where function $\theta(\cdot)$ is implemented by $1 \times 1$ convolutional layer, $C$ and $D$ represent feature map channels and number of nodes respectively. Then we can get the new features with the cross relationship between the source pose and the target pose.

$$V = H_{\text{source}} \cdot H_{\text{target}} \tag{5}$$

After obtaining new features, we use graph convolution for interactive reasoning. In particular, let $A \in \mathbb{R}^{D \times D}$ denote the node fully-connected adjacency matrix for spreading information across nodes, and let $W \in \mathbb{R}^{D \times D}$ denote the state update function. Identity matrix $I$ reduces the difficulty of optimization the graph convolution is formulated as:

$$Z = ((I - A)V)W \tag{6}$$

Following the principle in *Chen et al. (2019)*, Laplacian smoothing is used as the first step of volume product. Both $A$ and $W$ are adopt random initialization and updated by gradient descent. Next, we need to map the inferred $Z$ back to the coordinate space.

Similar to the first step, we adopt the projection matrix $H_{\text{target}}$ and linear projection $\varphi(\cdot)$ to formulate.

$$\widehat{F}^{P_c} = \varphi(H_{\text{target}} \cdot Z) \tag{7}$$

### *Texture block*

The texture blocks aims to transfer pose and texture simultaneously and interactively. Each texture blocks is consist of residual conv-blocks equipped with AdaIN. Firstly, we compute attention mask $M_t$ by two convolutional layers. Mathematically,

$$M_t = \sigma(Conv(\widehat{F}^{P_c})) \tag{8}$$

After getting the attention mask, the appearance code is updated by:

$$F_{sty}^i = F_{sty}^{i-1} M_t + F_{sty}^{i-1} \tag{9}$$

The pose code is updated by:

$$F^{P_c} = Conv(\widehat{F}^{P_c}) \parallel F_{sty}^i \tag{10}$$

where $\parallel$ means connecting along the depth axis.

## Encoder

The primary focus of the decoder is to generate a new image by decoding codes. We finally take the texture code to generate a new person image. According to standard practice, the decoder generates the generated image $I_g$ via $N$ deconvolutional layers.

## Discriminators

The main purpose of the discriminator is to promote the generator to generate a more realistic image by distinguishing the generated image from the real image. In the training process, we adopt pose discriminator $D_P$ and texture discriminator $D_t$ to identify the shape consistency and appearance consistency. The discriminators are implemented by Resnet Discriminator, each discriminator is independently trained, and all the discriminators can be analyzed and optimized separately.

## Loss function

### *Adversarial loss*

The goal of adversarial loss is to guide the images generated by the generator to be close to the real images. This goal is achieved by the min-max confrontation process between the generator and the discriminator. The discriminator needs to maximize the probability of correctly determining the distribution of real images and false image. The task of the generator is to identify minimize the probability of the generated images being identified as false images, the two continue to fight, and ultimately achieve Nash equilibrium. In this paper, an adversarial loss function with $D_P$ and $D_t$ is used to help the

generator optimize the generation parameters and synthesize the human body images in the target pose. The formula for adversarial loss in this paper is as follows:

$$\max_G \min_D \ L_{adv} = E_{I_t,I_c,P_t}\{\log[D_t(I_c, I_t) \cdot D_p(P_t, I_t)]\}$$
$$+ E_{I_t,I_g,P_t}\{\log[(1 - D_t(I_c, I_g)) \cdot (1 - D_p(P_t, I_g))]\} \tag{11}$$

where $\log[D_t(I_c, I_t) \cdot D_p(P_t, I_t)]$ represents the probability that the discriminator will distinguish the real image as real data. $\log[(1 - D_t(I_c, I_g)) \cdot (1 - D_p(P_t, I_g))]$ represents the probability that the discriminator will judge the generated image as a false image.

### Reconstruction loss

The goal of reconstruction loss is to improve the similarity between the original image and the generated images, avoid significant distortion of colors, and accelerate the convergence process. This paper uses L1 reconstruction loss to calculate the pixel difference between the generated source image $\hat{I}_c$ and the source image $I_c$. The formula is as follows:

$$L_{\text{pixel}-\text{rec}} = ||I_g - I_c||_1 \tag{12}$$

### Perceptual loss

Because we often use MSE loss function, the output images will be smoother (losing the details/high frequency part), so we can enhance the image details by choosing the perceptual loss function. The preceputal loss is computed as (*Ma et al., 2018*):

$$L_{per} = \frac{1}{W_jH_jC_j}\sum_{x=1}^{W_j}\sum_{y=1}^{H_j}\sum_{z=1}^{C_j}||\phi_j(I_g)_{x,y,z} - \phi_j(I_t)_{x,y,z}||_1 \tag{13}$$

where $\phi_j$ is the output feature of the $j$-th layer in the VGG19 network, and $W_j, H_j, C_j$ are the spatial width, height and depth of $\phi_j$, respectively.

The total loss function is denoted as:

$$L_{total} = L_{adv} + \lambda_r L_{pixel-rec} + \lambda_p L_{per} \tag{14}$$

where $\lambda_r$ and $\lambda_p$ denote the reconstruction losses and perceptual loss.

### Datasets and details

In this paper, we use DeepFashion (*Liu et al., 2016*) dataset for performance evaluation. DeepFashion contains 52,712 images with the resolution of $256 \times 256$. Before training, we use Human Pose Estimator (HPE) to remove noisy images from the dataset in which human body can't be detected by HPE. Here we select 37,258 images for training and 12,000 images for testing. In particular, the test sets do not contain the person identities in the training sets in order to objectively evaluate the generalization ability of the network. In addition, we implement the proposed framework in Pytorch framework using two NVIDIA Quadro P4000 GPUs with 16GB memory. The generator contains 9 cascaded residual blocks. To optimize the network parameters, we adopt Rectified Adam (RAdam), which can not only have the advantages of Adam's fast convergence but also possess the

advantages of SGD. We train our network for about 120k iterations. The learning rate is initially set $1 \times e-5$ and linearly decayed to zero after 60k iterations. The batch size for DeepFashion is set 1. We alternatively train the generator and discriminator with the above configuration.

### Metrics

Inception score (IS) (*Barratt & Sharma, 2018*; *Salimans et al., 2016*) and Structure Similarity (SSIM) (*Wang et al., 2004*) are the most commonly used indicators to evaluate the quality of generated images. Inception score uses the Inception Net V3 network to evaluate the quality of the generated images from two aspects: image clarity and diversity. Structure Similarity is a perception-based calculation model that measures the similarity of two images from three aspects: brightness, contrast, and structure. However, IS only relying on the generated image itself for judgment, ignoring the consistency between the generated image and the real image. What's more, based on this, Fréchet Inception Distance (FID) (*Heusel et al., 2017*) is adopted to measure the realism of the generated image. This method first converts both the generated image and the real image into a feature space, and then calculates the Wasserstein-2 distance between the two images. In addition to the above-mentioned objective evaluation indicators, a User Study was also conducted, and subjective indicators were formed by collecting volunteers' evaluation of the generated images.

## RESULTS AND DISCUSSION

### Quantitative and qualitative comparison

Since the judgment of the generated image is more subjective, We compare our method with several stare-of-the-art methods including PATN (*Zhu et al., 2019*), ADGAN (*Men et al., 2020*), PISE (*Zhang et al., 2021*). The qualitative comparison results are shown in Fig. 4. In terms of visual effects, our method achieves excellent performance. Our method avoids a lot of noise, such as the images in the first line of the figure, and other methods appear white noise points on the clothes, but our generated image has no noise and perfectly presents the style of the clothes in the source image. In addition, our method shows better details than other methods in hair and face, and is closer to the real image. For more details, zoom in on Fig. 4.

In order to verify the effectiveness of our method, we conducted experiments on four benchmarks. In order to get a more fair comparison, we reproduce PATN ADGAN PISE and test it with the test set in this paper. The results of comparison and the advantages of this method are clearly shown in Table 1. Our method is superior to other methods in SSIM and mask SSIM, which verifies the effectiveness of the Graph-based generative adversarial network and maintains the consistency of the structure in the pose conversion process. Although the IS value is slightly lower than that of ADGAN, the FID value is comparable, indicating that our generated images are very close to the real images.

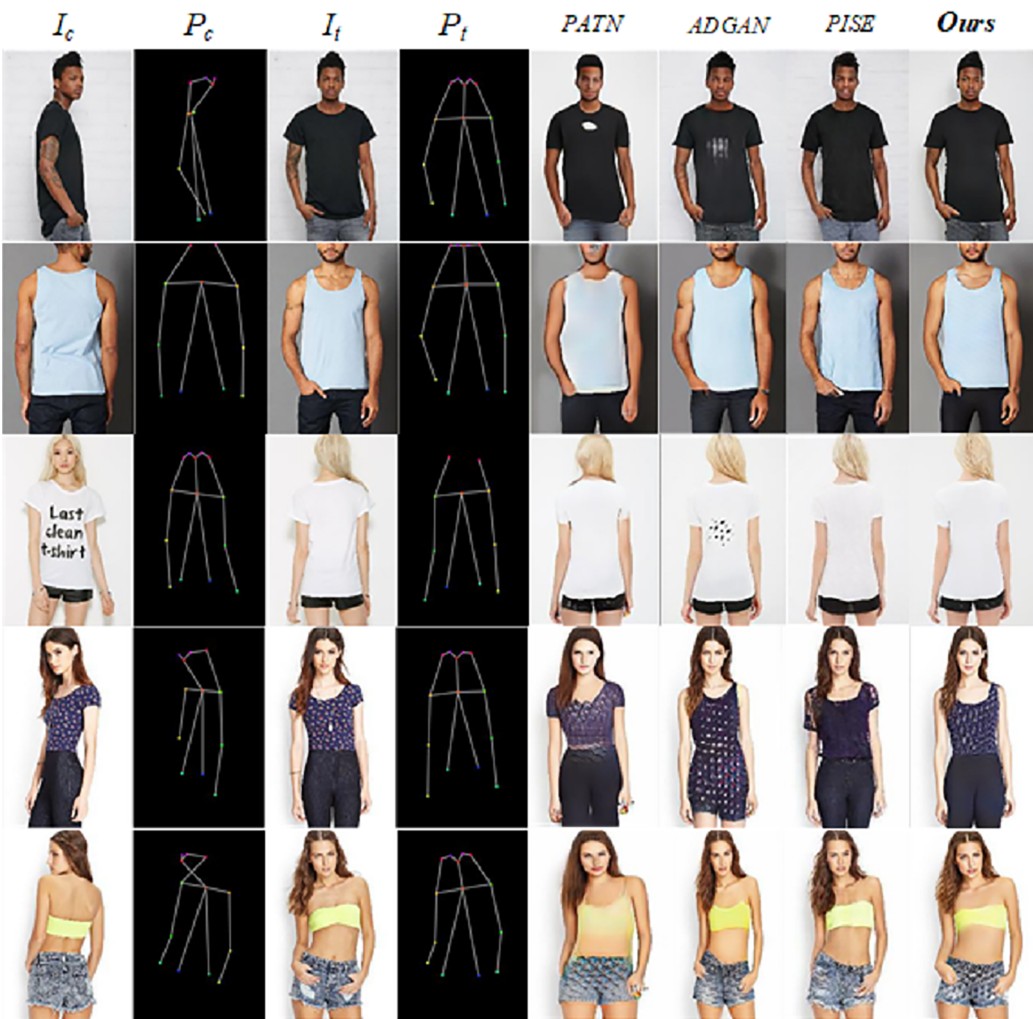

**Figure 4 Qualitative comparison with the state-of-the-art methods on DeepFashion.** Our results are shown in the last column.                

**Table 1 Quantitative comparisons with the state-of-the-art methods on DeepFashion.**

| Model | DeepFashion | | | |
|---|---|---|---|---|
| | IS | SSIM | Mask-SSIM | FID |
| PATN | 3.209 | 0.774 | / | / |
| ADGAN | 3.364 | 0.772 | / | / |
| PISE | / | / | / | 13.61 |
| PATN* | 3.054 | 0.7748 | 0.9275 | 20.374 |
| ADGAN* | 3.196 | 0.7736 | 0.9267 | 13.457 |
| PISE* | 3.233 | 0.7776 | 0.9281 | 13.286 |
| Ours | 3.1825 | 0.7916 | 0.9328 | 12.649 |

Note:
* The results tested on our test set.

**Table 2 User study results.**

| Indicator | DeepFashion | | | |
| --- | --- | --- | --- | --- |
| | PATN | ADGAN | Def-GAN | Ours |
| R2G | 19.14 | 23.49 | 12.42 | 22.84 |
| G2R | 31.78 | 38.67 | 24.61 | 39.45 |

## User study

Human subjective judgment is a very important indicator for generating images. This article relies on the questionnaire star website to do a difference test. In the experiment, 100 volunteers were asked to select the more realistic image from the generated images and the real images within one second. In order to ensure the confidence, following the rules in *Ma et al. (2018)*, we randomly select 55 real images and 55 generated images for out-of-order processing, and then pick out 10 of them for volunteer practice, and the remaining 100 for evaluation and judgment. Each image was compared 3 times by different volunteers. The results are shown in Table 2. The images generated by the method in this paper have achieved significant effects in human subjective evaluation.

R2G means the percentage of real images being rated as the generated w.r.t. all real images. G2R means the percentage of generated images rated as the real w.r.t. all generated images. The results of other methods are drawn from their papers.

## Ablation study

As shown in Fig. 5 and Table 3, the evaluation results of different versions of our proposed method are shown. We first compare the results using appearance decomposition to the results without using it. We remove the appearance decomposition part from the model, use the encoder similar to the PATN to encode the source image directly, and then transfer it to the generation network directly. By comparison, we find that the appearance decomposition module in our method can effectively improve the performance of the generator. It describes the spatial layout of the region level through the partition mapping, so as to guide the image generation with higher-level structural constraints. Then, we verify the role of graph-based global reasoning. In the pose pathway, we replace the graph-based global reasoning with the method used in *Zhu et al. (2019)*, which use the super position of convolution layer to expand the receptive field gradually for pose transfer. From the Table 3, the graph-based global reasoning module can get higher SSIM value, which shows that the module can improve the structural consistency of the image. In addition, we also verify the influence of each objective function on the generated results. It can be seen that adding these objective functions together can effectively improve the performance of the generator.
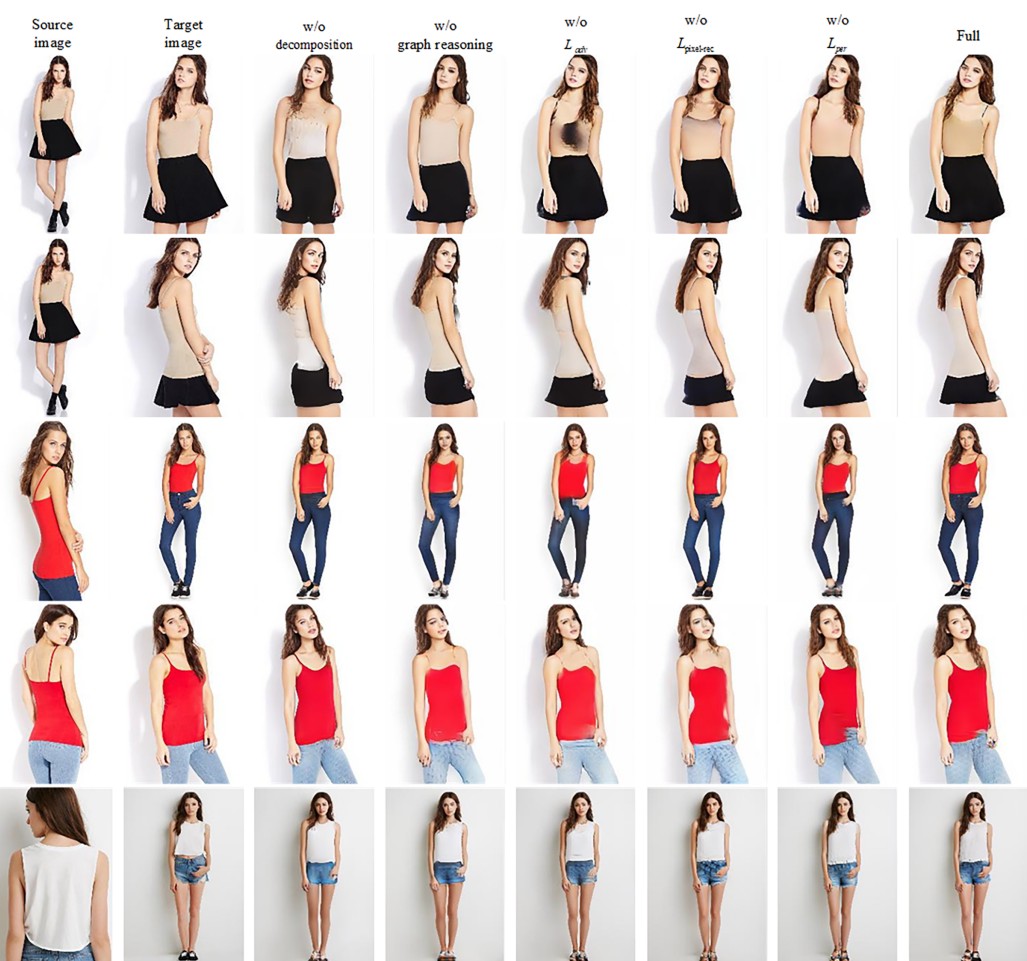

**Figure 5** The qualitative results of ablation study.

**Table 3** The evaluation results of ablation study.

| Model | DeepFashion | | | |
|---|---|---|---|---|
| | IS | SSIM | Mask-SSIM | FID |
| w/o decomposition | 3.128 | 0.781 | 0.930 | 14.862 |
| w/o graph reasoning | 3.025 | 0.778 | 0.929 | 17.306 |
| w/o $L_{adv}$ | 3.168 | 0.776 | 0.932 | 13.394 |
| w/o $L_{pixel-rec}$ | 3.164 | 0.774 | 0.931 | 12.672 |
| w/o $L_{per}$ | 3.178 | 0.785 | 0.933 | 14.862 |
| Full | 3.183 | 0.7916 | 0.933 | 12.649 |

## CONCLUSION

In this paper, a generation model based on appearance decomposition and graph-based global reasoning is proposed for pose guided image generation. The task of pose transfer is divided into pose path and appearance path. We use the graph network for global reasoning and appearance decomposition for texture synthesis simultaneously.

Through several comparative experiments on the Deepfashion dataset, our model shows superior performance in terms of subjective visual authenticity and objective quantitative indicators.

### Funding

This work was supported by the National Natural Science Foundation of China (Nos. 61462038, 61562039 and 61966016), the Jiangxi Provincial Natural Science Foundation (No. 20212BAB212005), the Science and Technology Planning Project of Jiangxi Provincial Department of Education (Nos. GJJ190217, GJJ190180 and GJJ200428), and the Open Project Program of the State Key Lab of CAD & CG of Zhejiang University (No. A2029). The funders had no role in study design, data collection and analysis, decision to publish, or preparation of the manuscript.

### Grant Disclosures

The following grant information was disclosed by the authors:
National Natural Science Foundation of China: 61462038, 61562039 and 61966016.
Jiangxi Provincial Natural Science Foundation: 20212BAB212005.
Science and Technology Planning Project of Jiangxi Provincial Department of Education: GJJ190217, GJJ190180 and GJJ200428.
Open Project Program of the State Key Lab of CAD & CG of Zhejiang University: A2029.

### Competing Interests

The authors declare that they have no competing interests.

### Author Contributions

- Yuling He conceived and designed the experiments, performed the experiments, performed the computation work, prepared figures and/or tables, and approved the final draft.
- Yingding Zhao conceived and designed the experiments, analyzed the data, authored or reviewed drafts of the paper, and approved the final draft.
- Wenji Yang conceived and designed the experiments, performed the experiments, authored or reviewed drafts of the paper, and approved the final draft.
- Yilu Xu conceived and designed the experiments, analyzed the data, prepared figures and/or tables, authored or reviewed drafts of the paper, and approved the final draft.

### Data Availability

The data is available at GitHub:
https://github.com/heyuling1019/person-image-generation.

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
