# Peer review of "Person image generation through graph-based and appearance-decomposed generative adversarial network"

_PeerJ Computer Science, doi:10.7717/peerj-cs.761_

## Round 0.1 · original submission · Minor Revisions

The authors should take the reviewers' comments into consideration in order to improve the overall quality of the paper.

Reviewer 1 ·

Basic reporting

There is a misspelling .....Apperance..even in the Title.
It should be taken care of before submission.

Experimental design

No comment

Validity of the findings

No comment

Additional comments

General Comments
This paper is concerned with personal image generation through graph-based networks along with decomposed particular attributes.
The authors put efforts to propose a generative adversarial network based on the graph.
Both qualitative and quantitative experiments were performed to confirm the proposed method.
The paper is interesting and will be beneficial to a sizable amount of researchers and students.
However the tying error even in the title of the paper ……Apperance …..
should be taken care of before submission.
Such errors are also seen in the text (For example Line 120)
Particular Comments.
1. Line 78 and Lines 93-94, the two statements about the proposed method may confuse the readers. Are they the same or different contributions?
2. Lines 147-148 Can we get the 18 joint points automatically?
3. I wonder how we can know the probabilities defined in equation 11 are non-zero.
The experimental results seem to be significant.

·

Basic reporting

The authors should check the manuscript once again by themselves.
- First of all, there are spelling mistakes in the title, and the capitalization rules are not consistent.
- The single-byte space after the commas and periods are not consistent.
- The operators that are supposed to be Hadamard products are written as e.
- Section 3.1 and 3.2 have the same titles. Perhaps 3.1 should be called "Encoder".

Experimental design

The method is written in a relatively clear manner, including mathematical expressions, but it does not correspond to Figure 3.
- For example, there is no mention of AdaIN, which is included in the Texture Block in the figure.
- Also, the text says that the Pose Block contains GCN, but the Pose Block in the figure is not drawn in such a way as to show this.

There is no mention of how the three Losses are combined.
If the three are to be added together, it should be clearly stated as such.

Validity of the findings

no comment

---

## Round 0.2 · accepted · Accept

Based on the reviewer's comments, the paper is now in good shape to be published.

Reviewer 1 ·

Basic reporting

No comment

Experimental design

No comment

Validity of the findings

No comment

·

Basic reporting

no comment.

Experimental design

no comment.

Validity of the findings

no comment.